# GASP: Cheap Uncertainty Quantification with Logic Neural Networks

**Liv Våge**
Princeton University
liv.helen.vage@cern.ch

**Lino Gerlach**
Princeton University
lino.oscar.gerlach@cern.ch

**Peter Elmer**
Princeton University
peter.elmer@cern.ch

## Abstract

Machine learning models are increasingly deployed where knowing when to trust them matters, yet uncertainty quantification (UQ) is often avoided since it can be costly to run. We show that this cost reduces dramatically for fast architectures which reduce to binary computations, such as logic neural networks. Replacing floating-point arithmetic with boolean gates makes inference fast and exposes the internal statistics needed for UQ almost for free. We introduce GASP (Gate-Activation Surprise Profile), a monitor derived from any trained logic neural network. GASP measures how improbable a gate's activation is relative to its training-time firing rate, averaged over an informative subset of gates, and from this primitive delivers selective prediction, and near- and far-OOD detection. Along with a simple calibration method, we cover a full UQ stack. Across four datasets—particle-physics jet tagging, network intrusion, ECG arrhythmia, and image recognition—GASP achieves strong performance on selective prediction, calibration and far-OOD detection at up to three to five orders of magnitude less compute than other algorithms, ceding only the near-OOD tier that requires gate covariance. With a very low added latency, this opens the door to cheap and deployable UQ in risk-critical areas.

## 1 Introduction

Models are arguably only as good as their uncertainties, yet uncertainty quantification (UQ) remains a sparsely used tool, even in risk critical domains. An illustration of this emerged during the COVID-19 pandemic, which saw a very dedicated ML effort. A metastudy of 400+ models for CT and CXR scans to diagnose and prognose Covid-19 found that *none* of the models were suitable for clinical use, many directly due to their unquantified uncertainties, biases and failure modes [1]. One large reason UQ remains underutilised is that there is no accepted SOTA methodology [2]. There are dozens of methods that work with certain datasets and certain conditions, leading to a large barrier to entry [3]. Even if this barrier is overcome, some of the best methods require a computational cost several times that of the inference of the models themselves, as we will see in subsection 2.1.

We argue that neural networks which reduce to binary computations, such as logic neural networks, can deliver much cheaper UQ. By counting the binary outputs at train time, UQ becomes cheap enough to run in a regime where it is usually much too expensive, like real-time FPGA inference. We thus introduce **GASP** (Gate-Activation Surprise Profile), a monitor that reads these gate statistics to deliver:

- **A complete UQ stack:** selective prediction, and near- and far- out of distribution (OOD) detection from one monitor, with a simple calibration method added - evaluated across four datasets. As

shown in Figure 1, it performs well in most tasks with a computational cost at times $10^3 - 10^5$ lower than the best performing alternative methods.

- **Anomaly detection almost for free:** While dedicated anomaly detection will usually perform better, we demonstrate that we can recover a lot of this signal in OOD while doing a related task, e.g. classification.

- **FPGA-native and cheap:** implementable in FPGA without DSP usage, at an estimated added latency of at around 10% or less for our experiments.

While this paper will focus on neural nets and classification, most of the GASP mechanism is transferrable to regression and to other logic gate ML methods, like decision tree based methods.

## 2 Background

### 2.1 Uncertainty quantification

Uncertainty in ML can come from two main sources: the data and the model. Irreducible noise in the data is *aleatoric*; uncertainty from the model's limited knowledge is *epistemic*, and is what we address. UQ methods serve one or more of three purposes:

- **Calibrated confidence**: deliver a predicted confidence that matches the observed accuracy.

- **Selective prediction**: flag likely misclassifications and abstain from predicting when relevant.

- **Out of distribution / drift detection**: flag input that is unlike the training data.

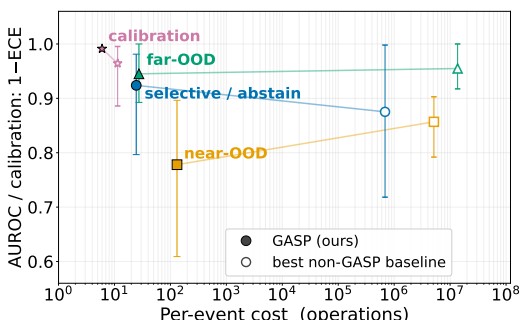

Figure 1: The performance of GASP vs. the best performing standard algorithms as measured across four datasets. GASP achieves equal or close to equal performance in three out of four categories at a fraction of the computational cost.

Each of these facets has mature methods that come at different costs. Calibrated confidence is nearly free. While prediction confidences are known to often be wrong, a recalibration like temperature scaling creates a baseline that is hard to beat [4, 5]. This baseline is also hard to beat when used as a selective prediction tool [6]. *Learning* mispredictions, e.g. via an extra class [7] or a dedicated head [8] often do well when there are ample misclassifications to learn from, and poorly otherwise [6].

OOD detection has many methods and no universal winner; the performance depends on the dataset and model [2]. The strongest fall into three families. *Feature-distance* methods [9, 10] score how far an input lands from the training data in representation space; they separate well but cost roughly $\mathcal{O}(d^2)$ in the feature dimension [11], or a stored bank of training features. *Sampling* methods, like ensembles [12] or dropout [13], treat disagreement across several predictions as uncertainty, at a multiple of the base inference cost. The computationally cheapest family reads the *activations* directly [14, 15], flagging inputs whose internal activation pattern departs from training at a very small cost.

### 2.2 Logic neural networks

A new family of neural networks is emerging that deliver fast inference by using only boolean operations - Logic Neural Networks (LNNs) [16–19]. We build on differentiable logic neural networks [19, 20], which learn to select logic gates at each position in the network. The training mechanism is largely irrelevant, however, since GASP operates directly on the boolean inference circuit. It is compatible with any inference mechanism where one can estimate the firing rates of the binary nodes, which also includes binary decision trees [21].

### 2.3 Related work

Since logic neural networks are relatively new, there is to the best of our knowledge no prior work that addresses their uncertainty quantification. Logic circuits are, however, long-established, and

a substantial literature estimates their reliability. Most relevant is signal probability [22], which is very established in circuit testing. It measures the probability that an output evaluates to *true* under a given input distribution. GASP also measures probabilities under an input distribution, but does so per input. The closest ML UQ method we are aware of is [23], which detects out-of-distribution inputs by binarising activation outputs. This is related in spirit, but is designed for continuous-valued DNNs rather than a natively boolean circuit.

## 3 Method

Our key insight to model UQ for LNNs comes from information theory. Each gate can output a 0 or 1, making it a Bernoulli distribution. The information $I$ gained by observing activation $a$ can be written as $I(a) = -log(q)$, where $q$ is the probability of observing that activation. This probability is something we can estimate directly from our training data. The surprisal $s_g$ of a gate $g$ is therefore

$$s_g(x) = -\big[a_g \log q_g + (1 - a_g) \log(1 - q_g)\big]. \tag{1}$$

Since $q_g$ is fixed after training, $s_g$ is a precomputed lookup per gate with no arithmetic at inference. We find that we do not need to calculate this for all the gates in the network, and doing so can dilute the signal. Instead, we can rank the gates by how informative they are, and average over $N$ gates.

**OOD and drift detection**. We posit that to identify data that is unlike training, our most informative signals come from the flipping of gates that are usually constant. We therefore rank our gates by fire-rate extremity $|logit(q_g)|$ and select $N$ sentinel gates. Our overall surprise score is then $S(x) = \frac{1}{N} \sum_g s_g(x)$. This catches inputs that differ from training globally, but is class-agnostic. To also catch inputs that look normal overall yet wrong for their predicted class, we calculate the firing probabilities of the gates within each class, $q_g^c$. We then recalculate the surprisals for the data in its predicted class. If it's atypical for the class it was assigned it signals both novel-class drift and a confident-but-wrong prediction that is worth abstaining on. We combine this global and class specific signal by standardising them on their training distributions and adding them together.

**Selective prediction**. We found that when there are few misclassification examples, simply using the confidence signal is best, and this is our anchor. When there are many examples, we train a simple classifier that learns when to abstain. With a small calibration dataset, we learn which gates fire differently when the model is wrong, and fit a small logistic regression $P(wrong) = \sigma(\sum_{g \in \mathcal{S}} w_g a_g + b)$.

**Confidence**. The easiest part of UQ for LNNs is measuring confidence. As is supported by the literature [6], we also find that no method reliably beats max probability. In our experiments we found that it was often poorly calibrated, so we recalibrate by fitting an isotonic function. This is a simple post-hoc metric that is easily stored in a 1D table. Binary activations combined with the confidence therefore give us an entire UQ stack by changing where in the logic network we consider our gates and how we compare them.

The tools that make up GASP will be released in *torchlogix* [24].

## 4 Results

Uncertainty quantification is task-dependent, so we evaluate across four datasets. As a canonical sanity check, we use **MNIST** [25] with FashionMNIST [26] as far-OOD. We then use jet tagging as a real-time science application. We use the hls4ml LHC jet dataset [27] as our in-distribution and another dataset, JetClass [28] as our OOD. Since the two jet tagging datasets are made from different physics generators, we use the same classes as near-OOD, representing detector drift or small physics changes. We use unseen classes (Higgs channels) as the far OOD. UQ is also highly relevant in medicine, so we apply GASP to **MIT-BIH** [29, 30] - an arrhythmia dataset - and use supraventricular ectopic beats (S) as near-OOD and unknown/paced beats (Q) as far-OOD. Another highly relevant application is security, so we use the network security dataset **NSL-KDD** [31]. This is a dataset normally used for anomaly detection, and we use it here to illustrate that a UQ algorithm can be used to obtain very cheap anomaly detection. We thus treat the novel attack subtypes absent from training as a single OOD tier. We follow the standard graded-OOD convention where near is a mild drift and far is strong; the MNIST/FashionMNIST and NSL-KDD splits are standard, while the MIT-BIH and

jet-tagging grading are our definitions. All results are shown in Table 1, evaluated on the inference graphs and averaged over three seeds.

As shown in Table 1, GASP achieves strong performance in three out of four UQ categories, at a fraction of the normal cost. Comparing our results of NSL-KDD to a SOTA anomaly detector like [32], we find that we are around 0.15 points lower. For a five way classifier where the anomaly detection comes almost for free, this is a meaningful performance. The weakest category is near-OOD. This is because GASP does not capture gate co-variance. A data sample may make the independent gates fire normally, but the joint structure may break - i.e. gates that usually fire together do not. Traditional methods like Mahalanobis [9] use the full covariance matrix, and DeepKNN [10] uses higher order relations. To estimate covariance a somewhat efficient way, we decompose the covariance matrix of the gate activations into eigenvectors, and keep $k$ as ordered

| Method | MIT-BIH | NSL-KDD | Jet tagging | MNIST | Ops/event |
|---|---|---|---|---|---|
| *Calibration — ECE* | | | | | |
| Max-prob (raw) | 0.233 | 0.422 | 0.020 | **0.005** | ~1 |
| Temperature scaling | 0.012 | 0.114 | 0.016 | **0.005** | ~1 |
| Dirichlet | 0.014 | 0.134 | **0.012** | 0.009 | ~1 |
| GASP (isotonic head) | **0.007** | **0.008** | 0.014 | 0.007 | ~1 |
| *Abstention — AUROC* | | | | | |
| Max-prob / temp-scaling | 0.847 | 0.727 | **0.797** | 0.959 | ~1 |
| DeepKNN | 0.693 | 0.723 | 0.602 | 0.737 | ~$10^5$ |
| Mahalanobis | 0.739 | 0.767 | 0.593 | 0.683 | ~$10^7$ |
| ConfidNet (off-circuit) | **0.994** | **0.998** | 0.718 | 0.790 | ~$10^6$ |
| GASP (anchored) | 0.959 | 0.981 | **0.797** | 0.959 | ~$10^2$ |
| *Near-OOD — AUROC* | | | | | |
| Max-prob / temp-scaling | 0.703 | 0.765 | 0.487 | – | ~1 |
| DeepKNN | **0.790** | **0.870** | 0.838 | – | ~$10^5$ |
| Mahalanobis | 0.753 | 0.864 | **0.907** | – | ~$10^7$ |
| GASP (sentinel+cc) | 0.609 | 0.829 | 0.896 | – | ~$10^2$ |
| *Far-OOD — AUROC* | | | | | |
| Max-prob / temp-scaling | 0.768 | – | 0.470 | 0.922 | ~1 |
| DeepKNN | 0.912 | – | 0.845 | 0.996 | ~$10^5$ |
| Mahalanobis | **0.946** | – | 0.924 | **1.000** | ~$10^7$ |
| GASP (sentinel+cc) | 0.893 | – | **0.943** | **1.000** | ~$10^2$ |

Table 1: Uncertainty quantification across four tasks on the inference graph, averaged over three seeds. Cost is arithmetic operations per event.

by the smallest informative axes, since the smallest variance axes usually carry a lot of OOD signal [33]. As Figure 2 shows, this recovers most of the OOD gap for several datasets. This is a known flaw with activation-based methods in UQ [33–35]. Though we know what causes the performance gap, we have not yet identified a way to make this computationally cheap. Even a $k = 32$ decomposition would cost $\mathcal{O}(10)$ times the model cost. We acknowledge this as a limitation, yet note that the effect is highly dataset dependent, and we still achieve a strong signal at a fraction of the normal cost.

On our smallest and largest networks—NSL-KDD ($\sim 1500$ gates) and MNIST ($\sim 16,000$ gates)—GASP adds an estimated few hundred to a thousand LUTs, roughly constant despite the order-of-magnitude size difference. The global surprise and selective-prediction legs run concurrently with the classifier and finish before the prediction. Only the class-conditional surprise score adds latency because it must wait for the argmax. It uses a small class-selecting multiplexer, adding a small amount of logic depth and an estimated $\approx 10\%$ of the classifier's latency.

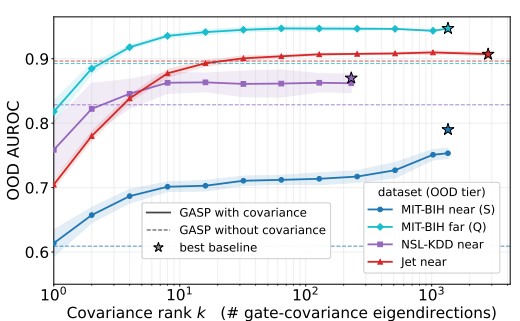

Figure 2: The performance of OOD with and without the inclusion of gate covariance. Including gate covariance with an eigendecomposition of the covariance matrix and keeping the top $k$ most relevant directions recovers most of the OOD performance gap for GASP, but at a high computational cost.

## 5    Conclusion

We presented GASP, an uncertainty monitor for logic neural networks that reads per-gate activations to cover calibration, selective prediction, and out-of-distribution detection at a fraction of the classifier's cost, with no retraining. On calibration, abstention and far-OOD it comes within roughly 5% of far more expensive methods while running three to five orders of magnitude cheaper for the latter two. It cedes only near-OOD, which we show requires gate covariance that we cannot cheaply reach. Through far-OOD, the same monitor also doubles as a cheap anomaly detector. By making uncertainty quantification close to free on a deployed circuit, GASP brings UQ to real-time FPGA inference, a regime where it has so far been largely absent.

This work was supported by the National Science Foundation under Cooperative Agreement PHY-2323298.

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
