# OpenReview forum: "GASP: Cheap Uncertainty Quantification with Logic Neural Networks"
_FastML/2026/Conference — FastML 2026 Conference Submission_

### Official Review · Reviewer_Jm6j · 2026-07-19
**Highly efficient uncertainty monitor for logic networks.**

**Rating:** 4
**Confidence:** 4

**Review:**

GASP proposes an uncertainty quantification (UQ) monitor for differentiable logic neural networks exploiting the fact that gate activations are Bernoulli-distributed and can be scored against their training-time firing rates at near-zero inference cost. The method covers calibration (isotonic recalibration), selective prediction (confidence anchor plus a logistic "abstention" model over informative gates), and OOD detection (a surprisal score over sentinel gates, optionally class-conditional). The paper is well-motivated as UQ is indeed often skipped in deployed ML due to cost, and LNNs' native boolean structure is a genuinely underexploited opportunity for near-free introspection. The related-work coverage is reasonable, and the paper is honest about its central limitation that near-OOD detection requires gate covariance the authors cannot yet compute cheaply.

Quality
The quality of this research is highly commendable. The authors identify the computational bottleneck of traditional uncertainty quantification and introduce an elegant solution for logic neural networks. By reading boolean gate statistics, the proposed Gate-Activation Surprise Profile (GASP) calculates how improbable an activation is compared to its training distribution. The empirical evaluation is robust, testing the framework on four distinct datasets: particle-physics jet tagging, ECG arrhythmia, network intrusion, and image recognition. The performance metrics accurately reflect the strengths and limitations of the approach.

Clarity
The manuscript is well structured and communicates its core concepts with excellent clarity. The authors seamlessly bridge information theory and hardware-level circuit logic. The distinction between global surprise scores and class-conditional surprise scores is well-explained. Furthermore, the explicit breakdown of performance across different UQ tasks calibration, selective prediction, near-out-of-distribution (OOD), and far-OOD, makes the results highly accessible to an interdisciplinary audience.

Originality
Applying information theory directly to the binary activations of logic neural networks to extract UQ is a highly original perspective. While the authors note that signal probability is established in classical circuit testing, repurposing this as a computationally cheap surprisal metric for deep learning models is a novel contribution. It successfully bypasses the need for costly feature-distance calculations or ensemble sampling.

Significance
This work holds substantial significance for the fast machine learning community. Delivering reliable UQ at three to five orders of magnitude less computational cost than competing baselines enables deployment in heavily constrained, risk-critical environments. The hardware implementation is highly efficient, utilizing a roughly constant number of look-up tables and adding an estimated latency of 10% or less.

Pros:
- Novel uncertainty quantification framework tailored to logic neural networks.
- Strong relevance to real time and FPGA based machine learning.
- Excellent computational efficiency compared with established UQ methods.
- Broad evaluation across multiple datasets and uncertainty tasks.
- Clear methodology and well motivated theoretical foundation.

Drawbacks:
- Near OOD detection performance lags leading baselines.
- Covariance based improvements reduce the claimed efficiency advantage.